# Are Saliva Tests Important in the Prediction of Carious Disease?

**Daniela Luminita Ichim [1], Liliana Sachelarie [1,*] , Gabriela Calin [1] and Alexandra Burlui [2]**

[1] Faculty of Medical Dentistry, "Apollonia" University of Iasi, 2 Muzicii Str., 700399 Iasi, Romania; danielaluminitaichim@yahoo.com (D.L.I.); m_gabriela2004@yahoo.com (G.C.)

[2] Department of Rheumatology and Physiotherapy, "Grigore T. Popa" University of Medicine and Pharmacy, 700115 Iaşi, Romania; alexandra.bului@yahoo.com

\* Correspondence: lisachero@yahoo.com

**Abstract:** (1) Background: The appearance and progression of carious lesions represent a complex phenomenon of interactions of microbial factors (the action of bacteria on the tooth), of the factors related to the host, to the diet, and to the time factor. Which hasan influence on the rate of microbismof the oral cavity on the installation of carious disease? (2) Methods: In order to correctly assess the cariogenic risk of an individual, it is recommended to perform twoor more tests based on different principles (microbiological, clinical, epidemiological). The representative data series for the investigation were analyzed statistically and by applying the Pearson correlation test considering the coefficient of determination R for all pairs of data series. (3) Results: Salivary tests played animportant role in establishing control sessions, in carrying out prophylactic caries therapy, and establishing prognosis. The existence of a statistical associationwas confirmed between the prevalence of dental caries and the results of salivary tests for the study group. (4) Conclusions: The results of the saliva tests can be used in oral health promotion.

**Keywords:** *Streptococcus mutans*; *lactobacilli*; salivary tests; dentistry; prevention; dental caries

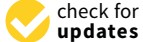



## 1. Introduction

Traditionally, the diagnosis and treatment of dental caries consist of detecting and restoring the cavity without considering the etiological factors.Even thoughstudies on the role of a particular bacterial species in initiating the carious process in humans are complicated by the fact that lesions can occur overmonths or years, there are sufficient arguments for assigning the essential role of mutagenic streptococcus cariogenesis such as *Streptococcus mutans* and *Streptococcus sobrinius*. There is a direct relationship between the number of streptococci and the development of the disease, their presence in large numbers in the saliva of individuals with active caries that arepresent in plaque as pioneer species [1–3]. An elevated level of mutant streptococci is present in most early lesions, but is also associated with the presence of persistent dental plaque. Another genus frequently involved in cariogenesis is *Lactobacillus*, which is a Gram-positive unsporulated, large Bacilli, cultivated on complex media [4–6]; it proliferates in the retentive areas and is responsible for the presence of the cavity, but can persist in large numbers after treating caries in the case of a diet rich in carbohydrates [7–9]. The involvement of other bacteria, such as *Actinomyces*, *Bifidobacterium*, *Eubacterium*, and even *Nocardia* species, is still insufficiently argued [2,7,9]. Both mutant streptococci and lactobacilli produce large amounts of acid by fermenting sugars from food consumed, initiating tooth decay by acid demineralization of the enamel [8,10,11]. In addition, *Streptococcus mutans* is a major producer of insoluble extracellular dextrans with which it adheres firmly to tooth surfaces; as a result of their metabolism, the environmental conditions are modified and become favorable for the development of other acidogenic bacteria [12–14]. The use of saliva tests (kits available on the market: CRT Bacteria, CRT Paraffin, Saliva Check Buffer) is expanding the view in clinical diagnosis, disease control, and decision for patient care and is useful for novel

ways for prediction, whilesaliva biomarkers are useful for many rapid tests or in laboratory procedures [15–17].

Salivary diagnostics is integrated as part of disease diagnosis; in this research, the association was confirmed between the prevalence of dental caries and the results of saliva test experimental samples [18]. The use of saliva tests in dental offices as a form of prophylaxis therapy can decrease the incidence of dental caries [18]. Qualitative diagnosis of saliva is superior for detecting the presence or absence of a marker compared to the quantitative diagnosis, which makes it an important means for detecting various pathologies.The microbiological composition, if modified, can increase the pathogenicity of the oral microbiome [19]. The oral microbiome is responsible for dental caries and periodontitis, two of the most common microbial-induced diseases worldwide. Some salivary parameters (secretion rate, consistency, viscosity, pH level) and buffer capacity are useful in defining the risk of oral diseases [19]. In this study, we assess caries activity by comparing twoor more tests based on different microbiological principles.

## 2. Materials and Methods

The study was conducted at the University ApolloniaofIasi on a group of 153 subjects (aged between 23–49 years, 97 women and 56 men)—volunteer patients under the coordination of the Department of Microbiology, Parasitology, and Virology, with the support of the Clinic of Cariology–Endodontics. All patients agreed to participate in the study. Saliva (unstimulated) was collected in the morning after oral hygiene procedures and collected in dedicated containers. Unstimulated and stimulated saliva samples were taken for each patient.Criteria for assessing the cariogenic activity wereanamnesis (eating habits), the clinical examination to determine the DMFT caries index, and the incidence of caries in the family.The predominant measure of dental caries is DMFT (decayed, missing and filled teeth). The saliva tests made use of commercial kits available on the market: CRT Bacteria, CRT Paraffin (IvoclarVivadent), Saliva Check Buffer (GC America Saliva Check Buffer).

The parameters considered for analysis were unstimulated (resting) saliva and the determined resting pH (5–5.8, high acidity; 6.0–6.6, moderate acidity; 6.8–7.8, normal salivary pH).

The series of data were analyzed by the application of the Pearson correlation test, on considering coefficient R, wherethecloser to zero, the weaker the correlation and is equal to the unit for a perfect correlation.

Stimulated saliva was obtained through the patient's chewing of a paraffin tablet for 5 min, followed by its collecting. The salivary indicators obtained are listed in Table 1.

**Table 1.** The salivary indicators for stimulated saliva and number of patients.

| Salivary Indicators | Values Obtained | Number of Patients | Normal Values |
|---|---|---|---|
| Stimulated salivary flux | 3.5–5 mL, low<br><3.5, very low | 132<br>11 | >5 mL, normal<br>(10 patiens) |
| Salivary secretion ratio | <1.1 mL/min, low | 112 | >1.1 mL/min, normal<br>(41 patients) |
| Salivary pH, Salivary buffer capacity | 6–9, low<br>0–5, very low | 105<br>32 | 10–12, normal<br>(16 patients) |

It is observed from Table 1 that a high percentage of people have low salivary indicators. Quantitative determination (in vitro) of *Streptococci mutans* and *lactobacilli* was made on selective culture media.

## 3. Results

Graphical analysis evidenced the variation of the salivary indicators as a function of the stimulated andsalivary flux.The observation to be made is that, at low values of the stimulated salivary flux, relatively high values of the DMFT parameter were registered,

which agrees with the idea that a salivary flow with a value over 5 mL may be correlated with a low caries risk (Figure 1). Similar evidence showed that DMFT takes relatively low values when the ratio of the stimulated salivary secretion is higher (Figure 2).

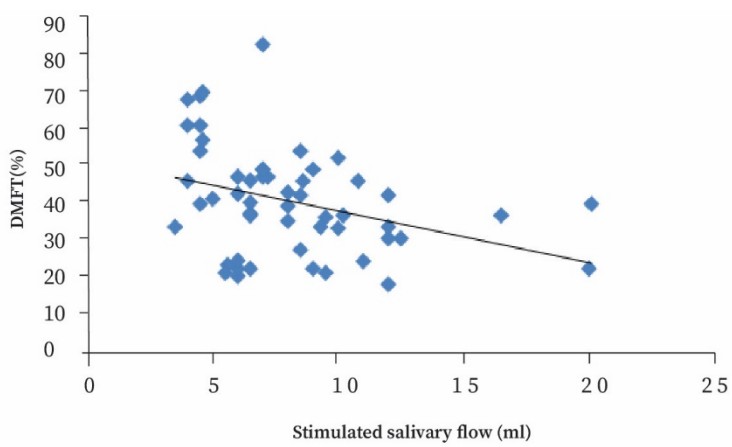

**Figure 1.** DMFT variation as a function of the stimulated salivary flux (mL).

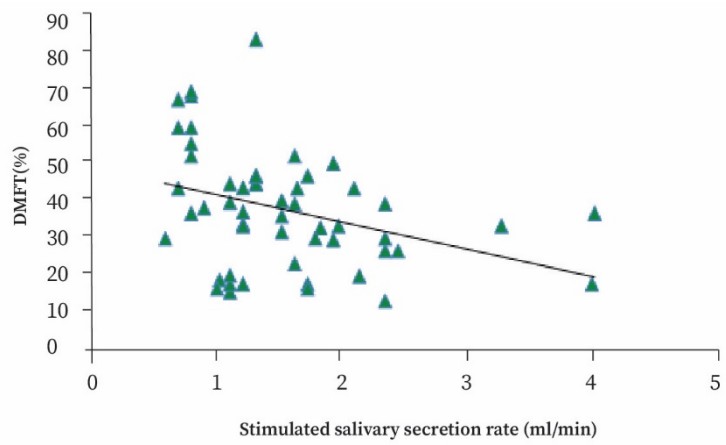

**Figure 2.** DMFT variation as a function of the ratio of the stimulated salivary flux (mL/min).

Figure 3 illustrates the dependence between salivary buffer capacity and stimulated salivary flow. The tendency of correlation by direct proportionality is obvious, even if almost the same value of the stimulated salivary flow is observed in several groups of subjects with different values of salivary buffer capacity, possibly because the salivary buffer capacity also varies depending on other parameters, such as salivary pH, DMFT, etc. The same observations are also valid for the dependence of the salivary buffer capacity on the ratio of stimulated salivary secretion (Figure 4)—which is quite normal, once it is known that a close linear correlation does exist between the stimulated salivary flux and the ratio of stimulated secretion (the two parameters differing with the same factor for all subjects).

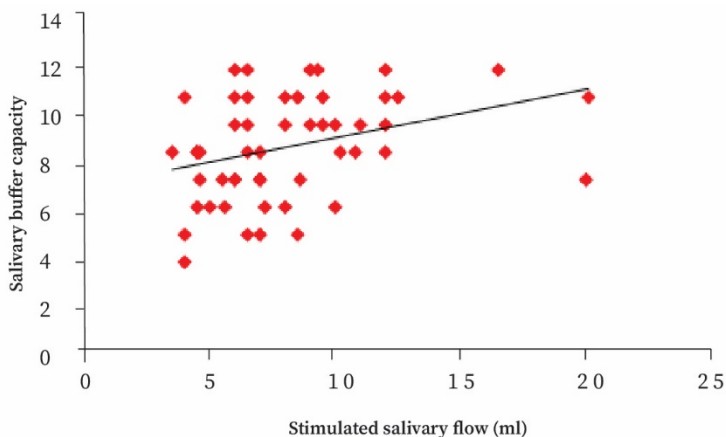

**Figure 3.** Dependence of salivary buffer capacity on the stimulated salivary flow (mL).

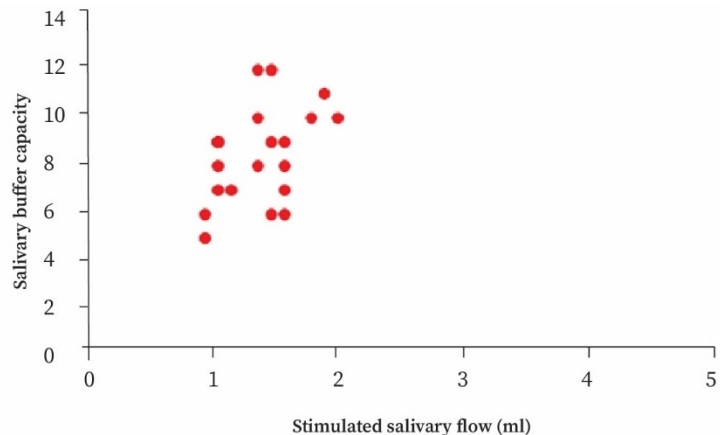

**Figure 4.** Dependence of the salivary buffer capacity on the ratio of stimulated salivary flow (mL/min).

The variation of the stimulated salivary flux in the function of pH is plotted in Figure 5. The observation to be made is that a salivary flow higher than 5 mL and a pH over 6.8 may be associated with a low caries risk. A similar discussion refers to the dependence of the ratio of salivary secretion stimulated by pH (Figure 6). In both cases, several subjects had the same pH value but different values of the salivary flux and of the ratio of salivary secretion.

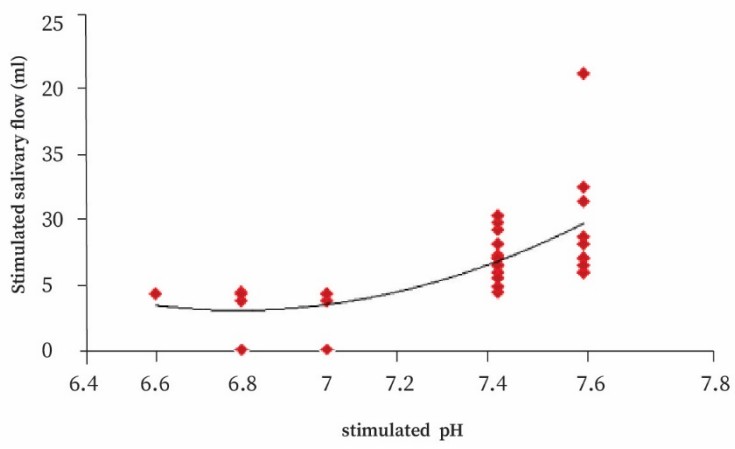

**Figure 5.** Dependence of the stimulated salivary flow (mL) on the stimulated pH.

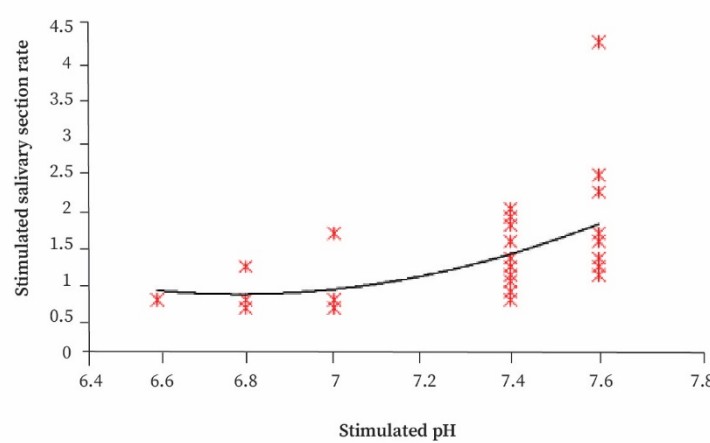

**Figure 6.** Dependence of the ratio of stimulated salivary secretion (mL/min) on the stimulated pH.

The linear correlation between DMTF and the unstimulated pH (Figure 7) is also very low, however, the graph suggests a statistical reduction of DMTF with the pH increase. Graphical representation between the two series of pH values (Figure 8) indicates that for each value of the stimulated pH, one or several values of the unstimulated pH are registered, even in the absence of a distinct mathematical correlation.

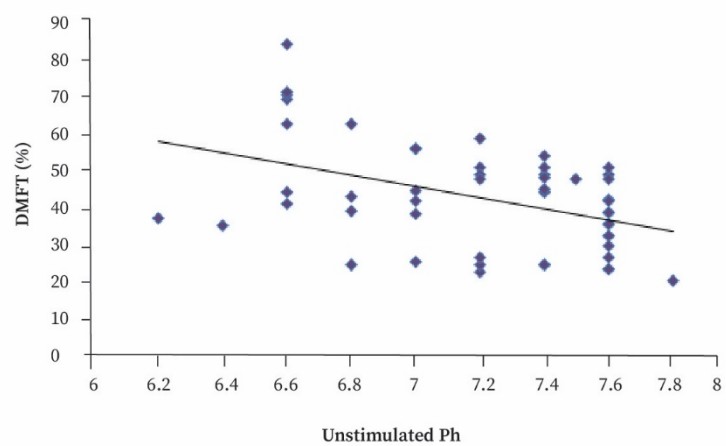

**Figure 7.** Correlation between DMFT (%) and unstimulated pH.

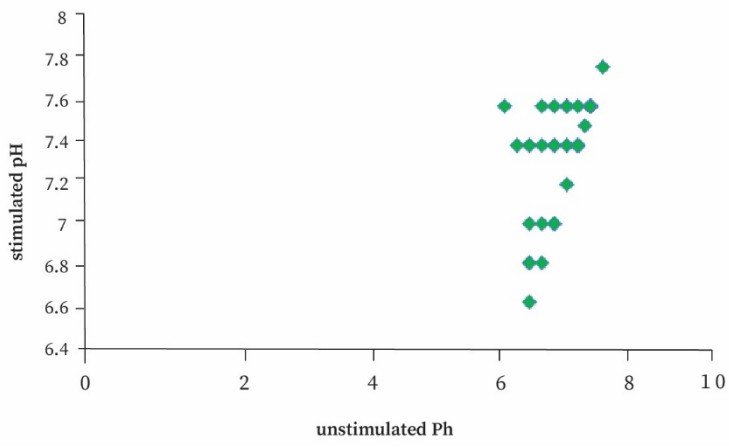

**Figure 8.** Correlation between stimulated and unstimulated pH.

It can be estimated that there is a certain tendency to increase the stimulation capacity with the increase of the stimulated pH—even if, practically, for each given value of the fivevalues that the stimulated pH takes, there are several different values of the stimulated capacity stimulation (Figure 9).

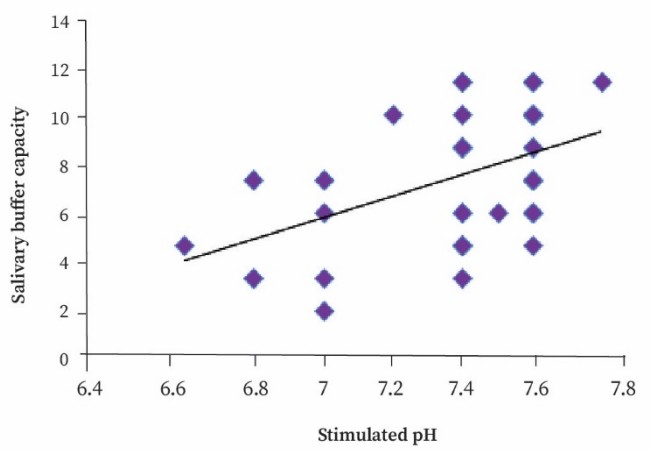

**Figure 9.** Correlation between salivary buffer capacity and stimulated pH.

The series of data representative for our investigation were analyzed both statistically and by the application of the Pearson correlation test, on considering coefficient R (which whencloser to zero the weaker the correlation is, and equal to the unit for a perfect correlation) for all pairs of data series; the situation is exemplified with the following cases:

✔ DMFT/pH—the relation between the two variables was rather weak (R = −0.4324);
✔ stimulated pH/stimulated salivary buffer capacity—again, the relation between the two variables was rather weak (R = 0.4111);
✔ unstimulated pH/stimulated pH—once again, the relation between the two variables was rather weak (R = 0.4877).

The histograms corresponding to the DMFT and salivary buffer capacity parameters (Figure 10) are similar to the graph of their normal distribution, with themaximum situated approximatively symmetrically to the limits of the respective domains of values in time, the histograms for the series of data representing the unstimulated and stimulated pH, respectively, being highly assymetrical in shape, with a maximum at the limit of the highest values from the respective series (mention being made of the fact that most of the subjects showed pH values over 7).

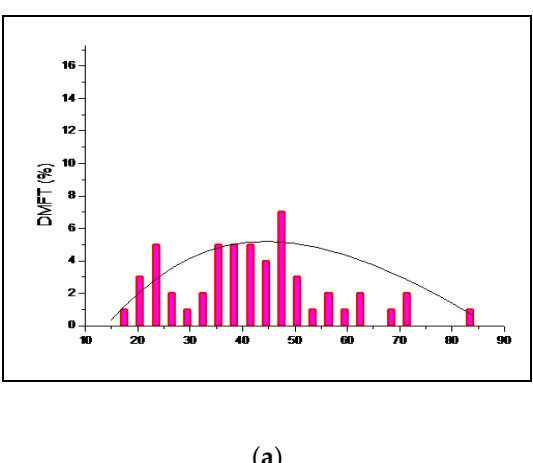

(**a**)

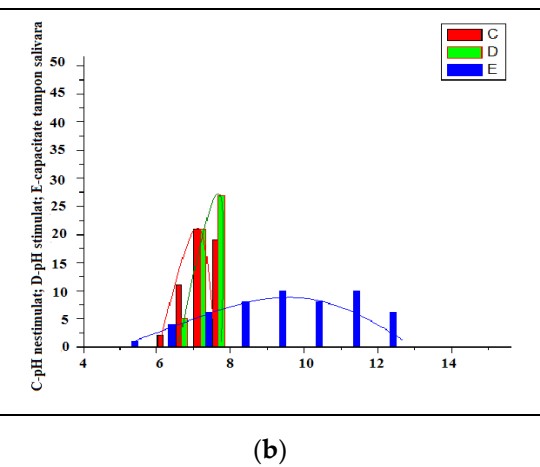

(**b**)

**Figure 10.** Histograms of the distributions of values for parameters: (**a**) DMFT (%); (**b**) unstimulated pH (C), stimulated pH (D), salivary buffer capacity (E).

As to the salivary buffer capacity, values dispersion is somehow higher, so that the corresponding box appears as having a reduced width and almost null wings (Figure 10) while, for the corresponding DMFT box, it is the largest—in the same manner in which the values from this series of data is (Figure 10a). In the case of nonstimulated and stimulated pH, the values hardly differ among them, which explains why the corresponding boxes are reduced, practically, to a single line (Figure 10b).

Thereis no correlation between the measured parameters and the age and/or sex of the subjects considered in the study. According to the recommendations of the manufacturer of kits for microbiological tests (CRT bacteria), the subjects with values < 105 CFU/mLsaliva, for both *Streptococimutans* and *lactobacilli*, have a low caries risk, unlikethose registering values >105—which have a high caries risk.

For an as correct as possible appreciation of the cariogenicrisk of any subject, the values of other parameters should be considered concomitantly, especially DMFT and the salivary buffer capacity (Figure 11), as well as the salivary pH and the stimulated salivary flow.

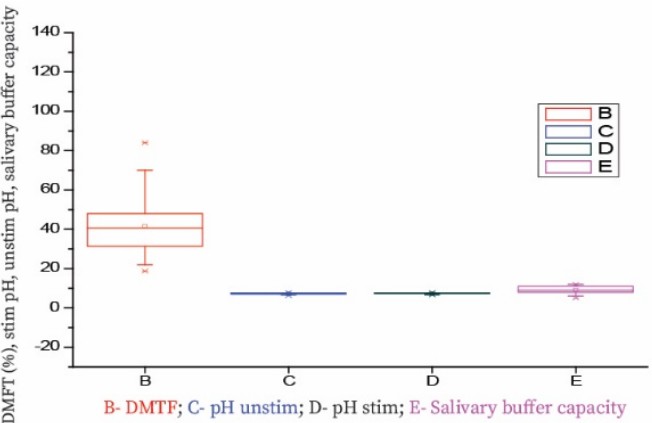

**Figure 11.** Box-chart representation of the four parameters.

## 4. Discussion

The certain characteristics of salivary factors and microbiological factors may help in dental caries prediction [20]. Some unexplored variables can have a significant influence on the oral environment [6]. The use of probiotics and natural compounds can modify clinical and microbiological parameters in periodontal patients, and they could have an effect also in response to the technique described in the present report [21]. For an as correct as possible appreciation of the cariogenic risk of any subject, the values of other parameters should be considered, such as DMFT, the salivary buffer capacity, salivary pH, the stimulated salivary flow, semi-quantitative determination (in vitro) of *streptococci mutans* and *lactobacilli*, as well as the possible correlations among them [22]. A stimulated salivary flux higher than 5 mLmay be correlated with a low caries risk, as it determines oral clearance of the microorganisms and of the alimentary rests, therefore a normal pH and a higher salivary buffer capacity. Equally, the salivary buffer capacity and the stimulated salivary flux are directly correlated with a low caries risk. Mention should be made of a certain increasing tendency of the salivary buffer capacity with the increase in the pH of stimulated saliva; a high value of salivary pH evidences a high salivary buffer capacity, as well. Analysis of the variation of the stimulated salivary flux as a function of pH leads to the conclusion that a salivary flux higher than 5 mLand a salivary pH over 6.8 is associated with a low caries risk. Salivary tests are performed to evaluate a number of parameters such as flow rate, buffering capacity, pH, viscosity, and salivary bacterial levels. Salivary tests have an important role in the development of prophylactic therapy and in establishing the prognosis. The results of salivary tests should be correlated with other tests such as diet or hygiene analysis. Only some of the parameters were analyzed in this study, but other parameters, such as saliva viscosity, can provide information in preventing caries.

## 5. Conclusions

The present study concludes that the saliva tests may help dentists play an important role in scheduling the control sessions, caries prophylactic therapy, and establishing the prognosis. Such tests may be employed toencourage patients to control their oral health. The study showed that it is important to monitor the patient's oral microbiotaand allows the intervention and prevention of oral disease.

These tests promote preventive dentistry, treat the cause, and eliminate risk factors in the development of caries. The results can be used as motivation in promoting the patient's oral health. Future studies on more specific salivary tests are needed.

**Author Contributions:** D.L.I.: conceptualization, investigation. L.S.: methodology, writing—reviewing and editing. G.C. and A.B.: software, validation. All authors have read and agreed to the published version of the manuscript.

**Funding:** This research received no external funding.

**Institutional Review Board Statement:** Not applicable.

**Informed Consent Statement:** Informed consent was obtained from all subjects involved in the study.

**Data Availability Statement:** Not applicable.

**Conflicts of Interest:** The authors declare no conflict of interest.

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
