# Peer review of "Are Saliva Tests Important in the Prediction of Carious Disease?"

_applsci, doi:10.3390/app11135932_

Round 1

Reviewer 1 Report

The Authors present the results of research on the parameters of saliva and their impact on the occurrence of caries. Unfortunately, most of the previous comments were not taken into account. Below is a list of them:

Lines 24-25 - There is still no justification for the research, and the sentence inserted at the beginning should be placed at the end of the introduction as a summary of the argument on the sense of the research. Adding it at the beginning has nothing to do with the next sentence. Besides, it is not justification, but information

Lines 46-49. The authors mention saliva tests, but do not specify exactly what tests are meant. Are these tests that quantify and qualitatively specify the bacteria present that contribute to caries? The publication is intended to reach a wide audience, which means that the names of the tests will not say anything to people from other countries where this type of test will not be available, so there must be information about what these tests detect. The authors still do not specify the biomarkers that can be taken into account in the diagnosis of saliva

Lines 64-66 - from the description it appears that the described parameters related only to unstimulated saliva? So what parameters were taken into account in the case of stimulated saliva? Lines 67-68 is only about obtaining it.

Table 1 - the title should contain information on which saliva this information relates to. There is still no information about what these values looked like in the study population. How many people have e.g. low salivary flux? Such information gives a picture, for example, what is the risk of caries in the studied population.

Figure 13, although it is a summary of the results, is unfortunately completely illegible.

Unfortunately, there is still no information on how the parameters of saliva changed in a given patient before and after stimulation. Diagrammatic representation of e.g. DMFT values before and after stimulation of salivation. Similarly, there is no tabulation of the obtained results.

Drawing axis descriptions remain unreadable.

The authors still do not provide information whether saliva was collected from each of the participants by both methods?

Author Response

The authors acknowledge the useful observations and suggestions of the reviewer’s as concerns the manuscript entitled “Are saliva tests important in the prediction of carious disease?”, co-authored by Daniela Luminita Ichim, Liliana Sachelarie, Alexandra Burlui

According to the reviewer’s recommendations, all the suggestions were taken into account, as follows:

Lines 24-25

The elements related to the introduction to the topic in introduction were removed and one phrase, i.e., “In this study we assess the caries activity by comparing 2 or more tests, based on different microbiological principles” was inserted in the end.

In order to better organize the work, a series of information was reorganized and critically selected, as follows:

Lines 46-49

 Salivary diagnostics is integrated as part of disease diagnosis and in this research the association was confirmed between the prevalence of dental caries and the results of the saliva tests experimental sample [18]. The use of saliva tests in dental offices as a form of prophylaxis therapy can decrease the incidence of dental caries [18]. Qualitative diagnosis of saliva is superior useful for detecting the presence or absence of a marker compared to the quantitative diagnosis, which makes it an important means for detecting various pathologies. The microbiological composition if have modification can increase the pathogenicity of the oral microbiome. Oral microbiome is responsible for dental caries and periodontitis, two of the most common microbial-induced diseases worldwide. Some salivary parameters (secretion rate, consistency, viscosity, pH level) and buffer capacity are useful in defining the risk of the oral diseases [19].

Lines 64-66- 67-68

Table 1

The saliva tests made use of commercial kits available on the market: CRT Bacteria, CRT Paraffin, Saliva Check Buffer.

The parameters considered for analysis are non-stimulated (resting) saliva and the determined resting pH (5-5.8 -high acidity, 6.0-6.6 – moderate acidity, 6.8-7.8 - normal salivary pH).

Table 1 The salivary indicators for stimulated saliva and number of patients

Salivary indicators

Values obtained

Number of patients

Normal values

 Stimulated salivary flux

3.5 –5 ml - low

132

> 5 ml – normal (10 patiens)

< 3.5 -  very low

11

 Salivary secretion ratio

< 1.1 ml / min – low

112

>1,1 ml / min – normal (41 patients)

Salivary pH – Salivary buffer capacity

6 – 9 - low

105

10 -12 – normal

(16 patiens)

0 – 5 - very low

32

It is observed from table 1 that a high percentage of people have low salivary indicators.

Figure 13

All figures were remade to be illegible.

Unstimulated and stimulated saliva samples were taken for each patient

I remain most respectfully yours,

Prof.dr. Liliana Sachelarie

Reviewer 2 Report

Dear authors,

Regarding the current state of the manuscript I didn't notice that my comments, from the first review, were considered much. At the current state I don't feel granting publication status would be recommended at least in my humble opinion. 

I'm sharing some typos and error I early found in the text:

Line 30 & 35: The authors are using the word Karyogenesis in the context of the starting of the demineralization of the enamel. I'm not sure if this is a translation issue or simply an error. Karyogenesis is the formation of the nucleus in a cell. The word should be cariogenesis. I'm sorry I completely missed this in the first revision though. 

Line 35: Fix "he is are gram-positive" e.g. "... Lactobacillus, which is a gram-positive ..."

Line 37: Authors are now referring to bacteria with a male gender pronoun. I suggest to change this to a neutral pronoun it instead. Don’t remember reading this in the first review, which is an unexpected and unusual change in the writing.

Line 39: Change italics in “as” to normal text

Line 55: Why Apollonia is within quotes? Is that necessary?

Line 73: Fix "Figure 1Culture media". I don't feel necessary to add these two figures since they are not even mentioned in the text and actually don't contribute much to the overall report.

In general the resolution of the figure is low and some of the text is too small which results in difficult reading and interpretation. As a side comment when creating plots I try to keep the font sizes consistent across all figures given they can be read easily and I save the images with a resolution no smaller than 300 dpi.

The formatting of the manuscript is not always consistent. e.g. After figure 6 line 96 has by eye the width of 3-4 rows.  After figure 12 line 134 has 1-2 rows width. After figure 13 line 143 has almost no separation with the previous line.

Regarding the references. Why the different margins?

Author Response

The authors acknowledge the useful observations and suggestions of the reviewer’s as concerns the manuscript entitled “Are saliva tests important in the prediction of carious disease?”, co-authored by Daniela Luminita Ichim, Liliana Sachelarie, Alexandra Burlui

According to the reviewer’s recommendations, all the suggestions were taken into account, as follows:

Line 30 & 35

An elevated level of mutant streptococci is present in most early lesions, but is also associated with the presence of persistent dental plaque. Another genus frequently involved in cariogenesis is Lacobacillus, which is a gram-positive, unsporulated, large bacilli, cultivated on complex media,  [3, 4, 6] who proliferate in the retentive areas, being responsible for the presence of the cavity, but he can persist in large numbers after treating caries in the case of a diet rich in carbohydrates [7,8,15].

Line 35-37

- issues were corrected and the word spacing issues were resolved.

Line 73 - the figure 1,2  was remove.

I remain most respectfully yours,

Prof.dr. Liliana Sachelarie

Round 2

Reviewer 2 Report

Dear Authors,

Thanks for the corrections.

Author Response

Dear reviewer,

The authors acknowledge the useful observations and suggestions of the reviewer’s as concerns the manuscript entitled Are saliva tests important in the prediction of carious disease?

by Daniela Luminita Ichim, Liliana Sachelarie, Gabriela Calin,Alexandra Burlui

According to the reviewer’s recommendations, all the suggestions were taken into account. In order to better organize the work, a series of information was reorganized and critically selected.

  1. I have done what you asked for and I apologize if I did not include everything from the first revision.
  2. I remade the images
  3. I revised the text
  4. I arranged the bibliography according to the requirement
  5. In the paper we did not intend to make a statistical analysis but rather to highlight the parameters taken in the study.

Thanks you very much!

I remain most respectfully yours ,

Prof. Liliana Sachelarie

This manuscript is a resubmission of an earlier submission. The following is a list of the peer review reports and author responses from that submission.

Round 1

Reviewer 1 Report

Dear authors,

The current state of the manuscript will require a significant amount of more work to reach a publication status. The manuscript is overpopulated with figures and without them, the manuscript narrative is really basic and runs extremely short. There is still merit in the idea, but it has to be properly tested in a appropriate statistical framework. The correlations are not strong and could be helpful to guide the group research decisions, but they are not definite prof that there is something really there. Alternative could be to use the data in a Machine Learning framework in order to determine if there is a discoverable trend in the data that can be used to make caries free/active status in patients. Following there are a few typos and comments related with the writing.

Line 13: Close parenthesis after “epidemiological”

Line 17: Add “space” after “establishing”

Line 28: Please follow the rules of presenting scientific names (Italics and correctly separated)

Line 35:  Add “space” after period in “CO2 [3,4,6].The”

Line 44: Italics and separate “Streptococcusmutans”. Never like this.

Reviewer 2 Report

The Authors present the results of research on the parameters of saliva and their impact on the occurrence of caries. There are numerous shortcomings in the work. Below is a list of comments:

  1. The presented introduction does not in any way refer to the presented research.
  2. In lines 47-50 the authors mention saliva tests, but do not specify exactly what tests are meant. They also do not specify biomarkers that can be taken into account in the diagnosis of saliva.
  3. Lines 64-68 are basically a repetition of lines 61-64, only slightly expanded.
  4. The work lacks demographic data of the participants
  5. The saliva collection procedure, the method of storing samples and the entire methodology for determinations are missing. Were both stimulated and unstimulated saliva collected from each patient? At what time, were there any additional restrictions, such as a break between a meal and saliva collection? Did the patient have to rinse the mouth additionally?
  6. Table 1 should also include information about patients. How many people had the value of a given parameter
  7. The results concern only the relationships that appeared in the case of stimulated saliva. The information in Chapter 2 shows that saliva was aspirated with and without stimulation. Data on unstimulated saliva may be more useful, especially since few people permanently chew paraffin.
  8. There is no information on how the parameters of saliva changed in a given patient before and after stimulation
  9. Descriptions of the drawing axes are illegible.
  10. There is no tabulation of the obtained results
  11. Figure 14 does not bring any information and its inclusion is incomprehensible. There is also no reference to this Figure in the text.
  12. Discussion is not a discussion. There is no reference to the current literature and the information contained therein represents information that is already known.
  13. I also have some doubts about the application. Hardly any or very little correlation has been shown, so how is this study supposed to help with prevention?